# Rural-urban disparities in nutritional status among ever-married women in Bangladesh: A Blinder-Oaxaca decomposition approach

**Md. Ismail Hossain**[1], **Md. Jakaria Habib**[2], **Faozia Afia Zinia**[2], **Azizur Rahman**[3], **Md Injamul Haq Methun**[4], **Iqramul Haq**[5]*

**1** Department of Computer Science and Engineering, Daffodil International University, Dhaka, Bangladesh, **2** Department of Statistics, Jagannath University, Dhaka, Bangladesh, **3** Department of Statistics, Jahangirnagar University, Savar, Dhaka, Bangladesh, **4** Statistics Discipline, Tejgaon College, Dhaka, Bangladesh, **5** Department of Agricultural Statistics, Sher-e-Bangla Agricultural University, Dhaka, Bangladesh

* iqramul.haq@sau.edu.bd

**Data Availability Statement:** In this study, we used data from the Bangladesh Demo-graphic

## Abstract

This study aims to investigate socioeconomic disparities in nutritional status among ever-married women in Bangladesh and to break down urban-rural differences in the underlying causes of undernutrition. We utilized data from the Bangladesh Demographic and Health Survey 2017–18, a sample size of 18328 ever-married women, including 5170 from urban residences, and 13159 from rural residences. To explore socioeconomic inequality, we employed a concentration indexing measure, while a multiple binary logistic regression model was carried out to identify the determinants associated with the outcome variable. A Blinder-Oaxaca decomposition analysis was performed to decompose the urban-rural gap in women's nutritional status using associated factors. The prevalence of undernutrition among ever-married women in Bangladesh was 12 percent. Notably, this percentage varied by region, with urban residents accounting for 8.6% and rural residents accounting for 13.3%. Our findings confirmed that undernutrition was more prevalent among women with lower wealth indexes in Bangladesh, as indicated by the concentration index (CIX = −0.26). The multivariable analysis investigating the determinants of undernutrition status among ever-married women, with a focus on residence revealed significant associations with respondent age, education, marital status, mass media access, wealth status, and division. According to the Blinder-Oaxaca decomposition and its extension, the prevalence was significantly higher in rural residences of Bangladesh than in urban residences, and the endowment effect explained 86 percent of the total urban-rural difference in undernutrition prevalence. The results of this study indicate that the factors that influence women's nutritional status in rural areas play a significant role in the gap, and the majority of the gap is caused by education and economic position. In order to effectively promote maternal health policies in Bangladesh, intervention techniques should be created that are aimed at the population, that is, the poorest and least educated.

Health Survey (BDHS), 2017-18, which is available from https://dhsprogram.com/data/available-datasets.cfm.

**Funding:** The author(s) received no specific funding for this work.

**Competing interests:** The authors have declared that no competing interests exist.

## 1. Introduction

Malnutrition among women remains a major public health problem in many parts of the world, specifically in low- and middle-income countries [1,2]. It can be classified as undernutrition, overweight, and obesity. It is well known that maternal nutrition is an important determinant of pregnancy outcomes. Maternal height and weight could affect intrauterine fetus growth before or after conception [2]. After birth, maternal nutritional status has a significant effect in determining child health outcomes [3]. Therefore, it is important to address the nutritional needs of married, pregnant, and breastfeeding women around the world.

Research into maternal malnutrition, particularly the impact of low nutritional status on maternal mortality and morbidity, remains crucial in developing countries for the foreseeable future [4]. It is worth noting that globally, 10 percent of women are underweight, 15 percent are obese [5], over 35 percent suffer from anemia [6]. These underweight or overweight/obese is a significant risk factor contributing to the global burden of diseases [7].

According to several studies, low nutritional status among women in the regions of south-central and south-east Asia and sub-Saharan Africa remains unacceptably high [3,8]. For example, the prevalence of a low body mass index was approximately 40% in sub-Saharan Africa [9]. Research conducted in nine African countries, including Ethiopia, found that more than 20 percent of women are underweight [3]. In South Asia, one in five women of reproductive age has a low body mass index [10]. Efforts have been made in Bangladesh to address the issue of malnourishment among reproductive-aged women. For example, recent studies based on Bangladesh Demographic and Health Survey have highlighted the prevalence of underweight and overweight women in Bangladesh, with estimates of 12 percent and 24 percent, respectively [11,12]. However, despite these findings, there remains a high prevalence of low nutritional status among rural Bangladeshi women of reproductive age, highlighting regional disparity [13]. This disparity has significant adverse health consequences for both women and their offspring, indicating that the overall prevalence of satisfactory nutritional status among women in Bangladesh is a matter of concern.

While researchers have attempted to determine the risk factors for low nutritional status among women using nationally representative data in the past, it is equally important to quantify the magnitude of socioeconomic inequalities in health and nutrition. The 2021 global hunger index ranks Bangladesh 76[th] out of 116 country, emphasizing the ongoing challenge of hunger and malnutrition within the country [14]. Despite significant progress in nutrition and health in the past decade, approximately a quarter of the total population of Bangladesh experienced food insecurity in 2019 [15,16].

The Bangladesh National Strategy for Maternal Health (BNSMH) 2017–2030's overarching goal is to provide direction to the Ministry of Health and Family Welfare (MOHFW) and the Government of Bangladesh (GoB) in addressing the gaps and inequities in the provision of high-quality maternal health services as well as the social and development factors that have an impact on maternal health [17]. According to Bangladesh constitution's Article 19(2), the state must take decisive action to eliminate social and economic disparities and achieve a consistent level of economic development throughout the country. This is a crucial issue that deserves the attention of our scholars and policymakers because it is a common goal in both rich and developing nations to achieve economic and physical equality. As part of its "inclusive economic growth" strategy in the Eighth Five-Year Plan, the government's development program should prioritize reducing spatial inequality. The government must focus on create equitable regional development on closing infrastructure gaps, developing human capital, enhancing primary health care and nutrition, and so on.

We acknowledge that previous studies have estimated the prevalence of undernutrition and examined socioeconomic inequalities among women of childbearing age in Bangladesh.

However, there are still gaps that need to be addressed. It is important to acknowledge these gaps and emphasize how our study contributes to filling them. Specifically, our research focuses on the regional disparities in nutritional status among women in Bangladesh. Along with this gap, our research aims to understand the extent to which the difference in mean outcome between urban and rural areas. To achieved this objective, we employ a Blinder-Oaxaca decomposition analysis which allows us to decompose the undernutrition gap into two components; one that is explained by differences in the level of the determinants, and another component that can be attributed to differences in the effect of these determinant on the nutritional status of women [18]. By conducting this decomposition analysis, our study provides valuable insights for policymakers, especially in the field of public health initiatives. It helps identify the socioeconomic gap in nutritional status between rural and urban areas in Bangladesh. Addressing this gap is crucial not only for achieving the goals set by the ruling government's manifesto but also for meeting national and international targets, such as Sustainable Development Goal 3 (SDG 3).

## 2. Materials and methods

### 2.1. Data source

A nationally representative secondary dataset was used for this study, titled "Bangladesh Demographic and Health Survey, 2017–18", which was implemented by the National Institute of Population Research and Training (NIPORT) and funded by the United States Agency for International Development (USAID).

### 2.2. Sample design

This cross-sectional survey used a two-stage stratified sampling design, where 675 enumeration areas were selected in the first stage, and then 30 households were selected from each enumeration area in the second stage. The survey was conducted in 20250 households and 20108 completed interviews with 15 to 49-aged women. Currently, pregnant women were omitted from this sample. Using weighting factors provided by the Bangladesh Demographic and Health Survey, the data were weighted to represent a more accurate structure of the Bangladeshi population for further analysis. After weighing, 18328 ever-married women were included in this study (5170 from urban residences and 13159 from rural residences in Bangladesh). Ever-married refers to women who have been married at least once in their lives, even though their current marital status may not be "married".

### 2.3. Dependent variable

The dependent variable for this study was "under nutritional status among ever married women", which was assessed based on body mass index (BMI) and defined by,

$$BMI = \frac{Weight\ in\ kg}{(Hight\ in\ meter)^2}$$

According to world health organization, women with BMI<18.5 are considered as undernutrition. This study recodes dependent variable as,

$$Under\ nutritional\ status = \begin{cases} 1, If\ BMI < 18.5\frac{kg}{m^2} \\ 0, Otherwise \end{cases}$$

## 2.4. Independent variables

The following socio-demographic and economic variables were included: Respondent age in years (15–19, 20–24, 25–29, 30–34, 35–39, 40–44, 45–49), Respondent education (No education, Primary, Secondary, Higher), Respondent working status (Yes, No), Marital status (Married, Others), Mass media access (Not exposed, Exposed), Wealth status (Poorest, Poorer, Middle, Richer, and Richest), Residence (Urban, Rural), Divisions (Barisal, Chattogram, Dhaka, Khulna, Mymensingh, Rajshahi, Rangpur, Sylhet). To capture access to media, we combined three separated variables: listening to the radio, watching television, and reading a newspaper. The authority of Bangladesh Demographic and Health Survey provides a wealth index indicator, which serves as a composite measure of a household's cumulative living standard. This indicator is calculated using information about a household's ownership of selected assets: televisions and bicycles; materials used for housing construction; and types of water access and sanitation facilities. For a detailed explanation of how the wealth score is calculated, please refer to the survey report available at "https://dhsprogram.com/publications/publication-FR344-DHS-Final-Reports.cfm".

## 2.5. Statistical methods

**2.5.1. Bivariate, and multivariable analysis.** A simple descriptive analysis, bivariate analysis, and multivariable analysis were conducted in this study. The descriptive analysis describes the percentage distribution of the variables. In bivariate analysis, to examine the association between under nutritional status among ever-married women and various independent variables, which were selected by studying previous literature, in this case, we have applied the $\chi^2$ test.

In the multivariable setup, the association between independent variables and the under nutritional status among ever-married women can be examined using binary logistic regression.

Let $M_i$ denote the binary dependent variable for the $i^{th}$ observation,

$$\text{where, } M_i = \begin{cases} 1, \text{If BMI of a woman is} < 18.5\text{kg/m}^2 \\ 0, \text{If BMI of a woman is} \geq 18.5\text{kg/m}^2 \end{cases}$$

$I_{i1}, \ldots, I_{ip}$ be a set of independent variables that can be quantitative or indicator variables referring to the level of categorical variables.

Since $M_i$ is a binary variable, it has a Bernoulli distribution with parameter $\pi_i$. The dependent of the probability of success on independent variables is assumed to be respectively as-

$$P(M = 1) = \pi_i = \frac{\exp(\beta_0 + \beta_1 I_{i1} + \cdots + \beta_p I_{ip})}{1 + \exp(\beta_0 + \beta_1 I_{i1} + \cdots + \beta_p I_{ip})}$$

The above relation also can be expressed as,

$$g(I) = logit(\pi_i) = log\frac{\pi_i}{1 - \pi_i} = \beta_0 + \beta_1 I_{i1} + \cdots + \beta_p I_{ip}$$

The odds ratio with a 95% confidence interval was usually used to examine the association of predictor variables with the low level of nutritional status.

**2.5.2. Inequality analysis.** This study includes an assessment of socio-economic inequality, utilizing the widely used "Concentration index" as a measure of inequality. It is defined as,

$$CIX = \frac{2cov(M_i|R_i)}{\bar{M}}$$

here CIX is the concentration index, $R_i$ be the fractional rank in the distribution of socioeconomic position, $M_i$ is the dependent variable index, and $\bar{M}$ is the mean of the outcome variable of the sample.

This study is trying to estimate the value of 'CIX' based on the Lorenz curve/concentration curve, and there is a gap between the concentration curve and the 45˚ line, indicating that there is no relationship between the two variables. The range of the value of CIX is −1 to +1, and this sign indicates the direction of any relationship between the health variable and socio-economic position.

**2.5.3. Decomposition analysis.** A popular decomposition analysis, named "Blinder-Oaxaca decomposition" was used to evaluate the residential variation of ever-married undernourished women in Bangladesh.

Let, $M_i$ be the binary dependent variable and $I_i$ is the set of independent variables, and there are two groups (urban and rural). Then, the following equations can be written,

$M_i^{Rural} = \beta^{Rural}I_i + \epsilon_i^{Rural}$; [if a woman from rural residence]

$M_i^{Urban} = \beta^{Urban}I_i + \epsilon_i^{Urban}$; [if a woman from urban residence]

Thus, the regional gap can be written as,

$$M^{Urban} - M^{Rural}$$

$$= (I^{Urban} - I^{Rural})\beta^{Rural} + (\beta^{Urban} - \beta^{Rural})I^{Urban} + (I^{Urban} - I^{Rural})(\beta^{Urban} - \beta^{Rural})$$

$$= \Delta I\beta^{Urban} + \Delta\beta I^{Rural} + \Delta I\Delta\beta$$

$$= E + C + CE$$

Here, the overall urban-rural gap in ever-married women undernutrition is comprised of the gap in the endowment effect (E), and the gap between the coefficients effect (C), and the interactions effect (CE).

## 2.6. Software

For data wrangling Statistical Package for the Social Sciences (SPSS) version 25 was used. STATA version 16 was used to perform the bivariate, multivariable analyses, inequality and decomposition analysis in this study.

## 3. Results

### 3.1. Association of undernutrition status among ever-married women by selected variables

Table 1 describes the background characteristics of the ever-married women who participated in the study. The highest number of respondents were from Dhaka (25.2 percent) divisions, as well as rural residences (71.8 percent). Almost half of the respondents were not involved in any kind of work (50.7 percent) and were exposed to the media (55.8 percent). Most participants (17 percent) were between 25–29 and 30–34 years of age. Approximately an equal

**Table 1. Percentage distribution of undernutrition status among ever-married women according to background characteristics.**

| Background characteristics | Frequency (%) | Undernutrition among ever-married women | | | |
|---|---|---|---|---|---|
| | | Urban | | Rural | |
| | | (%) | $\chi^2$ (p-value) | (%) | $\chi^2$ (p-value) |
| **Respondent age (in years)** | | | | | |
| 15–19 | 1620 (8.8) | 25.3 | 205.08 (<0.001) | 24.1 | 179.88 (<0.001) |
| 20–24 | 3006 (16.4) | 13.3 | | 15.9 | |
| 25–29 | 3182 (17.4) | 6.5 | | 11.6 | |
| 30–34 | 3217 (17.6) | 4.6 | | 10.9 | |
| 35–39 | 2805 (15.3) | 5.9 | | 9.5 | |
| 40–44 | 2247 (12.3) | 5.3 | | 12.4 | |
| 45–49 | 2251 (12.3) | 7.4 | | 12.8 | |
| **Respondent education** | | | | | |
| No education | 3220 (17.6) | 10.6 | 19.97 (<0.001) | 16.6 | 49.33 (<0.001) |
| Primary | 5815 (31.7) | 10.2 | | 14.1 | |
| Secondary | 7116 (38.8) | 8.2 | | 11.8 | |
| Higher | 2178 (11.9) | 5.6 | | 9.7 | |
| **Working status** | | | | | |
| Yes | 9035 (49.3) | 9.5 | 3.29 (0.04) | 14.1 | 9.85 (0.002) |
| No | 9293 (50.7) | 8.1 | | 12.3 | |
| **Marital status** | | | | | |
| Others | 1117 (6.1) | 10.9 | 3.33 (0.03) | 16.8 | 8.83 (0.002) |
| Married | 17212 (93.9) | 8.5 | | 13.0 | |
| **Mass media access** | | | | | |
| Not exposed | 8109 (44.3) | 14.1 | 63.10 (<0.001) | 16.5 | 132.51 (<0.001) |
| Exposed | 10219 (55.8) | 6.9 | | 9.7 | |
| **Wealth status** | | | | | |
| Poorest | 3408 (18.6) | 15.8 | 110.24 (<0.001) | 21.2 | 376.79 (<0.001) |
| Poorer | 3625 (19.8) | 14.4 | | 16.3 | |
| Middle | 3705 (20.2) | 13.0 | | 10.1 | |
| Richer | 3830 (20.9) | 10.6 | | 8.0 | |
| Richest | 3761 (20.5) | 4.5 | | 4.1 | |

*(Continued)*

**Table 1.** (Continued)

| Background characteristics | Frequency (%) | Undernutrition among ever-married women | | | |
|---|---|---|---|---|---|
| | | Urban | | Rural | |
| | | (%) | $\chi^2$ (p-value) | (%) | $\chi^2$ (p-value) |
| **Residence** | | | | | |
| Urban | 5170 (28.2) | - | - | - | - |
| Rural | 13159 (71.8) | - | - | - | - |
| **Divisions** | | | | | |
| Barisal | 1024 (5.6) | 7.7 | 20.89 (0.004) | 12.3 | 196.42 (<0.001) |
| Chattogram | 3248 (17.7) | 7.6 | | 7.7 | |
| Dhaka | 4611 (25.2) | 8.3 | | 11.4 | |
| Khulna | 2163 (11.8) | 8.9 | | 11.5 | |
| Mymensingh | 1401 (7.6) | 9.7 | | 19.8 | |
| Rajshahi | 2607 (14.2) | 8.2 | | 14.0 | |
| Rangpur | 2211 (12.1) | 9.2 | | 15.4 | |
| Sylhet | 1063 (5.8) | 17.4 | | 22.7 | |

number of participants (approximately 18 to 20%) belonged to the poorest, poorer, middle, richer, and richest wealth statuses. More than one-third (38.8 percent) of the respondents had completed secondary education.

The relationship between socio-demographic and economic characteristics and the undernutrition status among ever-married women in regional areas of Bangladesh is defined in Table 1. From the $\chi^2$ test all the covariates such as respondent age (in years), respondent educational status, respondent working status, marital status, mass media, wealth status, and division were found significantly associated with undernutrition among ever-married women in both urban and rural areas of Bangladesh ($P<0.001$; $P<0.01$; $P<0.05$).

Fig 1 illustrates the regional disparities in the prevalence of undernutrition among ever married women in Bangladesh. The data indicates that approximately 12% of ever-married women in Bangladesh were affected by undernourishment. Specifically, the prevalence of undernutrition was found to be 8.6% among women residing in urban areas and 13.3% among those in rural areas. The proportion test revealed a significant difference between urban and rural residences in Bangladesh ($\chi^2$ = 102.3, $P<0.001$).

## 3.2. Factors associated with ever-married women's undernutrition

Table 2 represent the binary logistic regression analysis of undernutrition status among ever-married women in urban and rural context. Table 2 shows that young women were significantly more likely to be undernourished than other age group of women, in both urban and rural areas. In rural area, primary educated women were 17 percent less likely to be undernourished than women who had no education, whereas higher educated women were 21% less

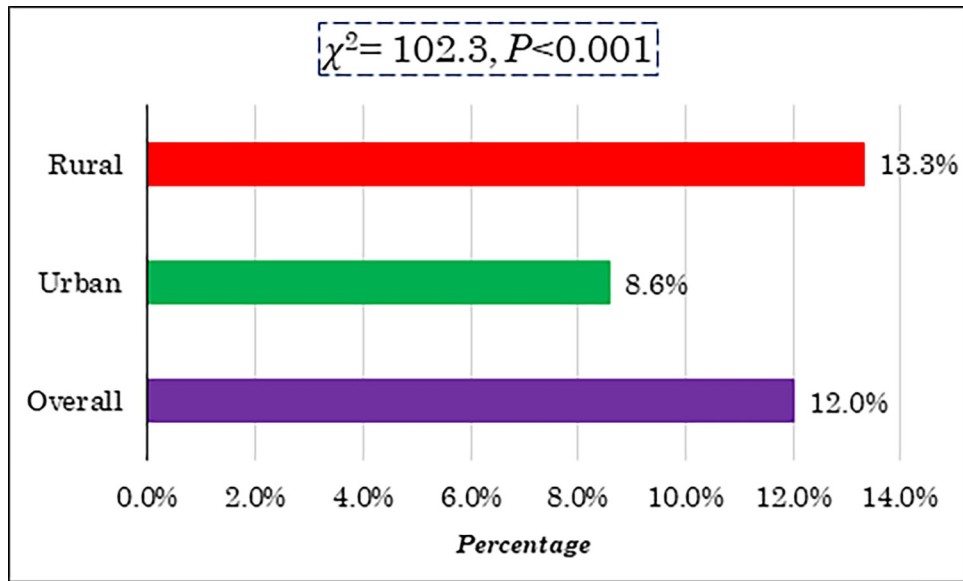

**Fig 1. Regional variations of ever-married undernourished women in Bangladesh.**

likely to be undernourished. On the other hand, in the urban area, highly educated women were 40% less likely to be undernourished. Women in the rural area, who were not working were 12% less likely to be undernourished. Married women in urban areas were 33% less likely to be undernourished, while in rural area married women were 19% less likely to be undernourished. Women who were exposed to mass media, were less likely to be undernourished. Women belonging to the poorest household were 4.60 times more likely to be undernourished than women from the richest household in rural areas, while in urban area, women with poorest household were 2.55 times more likely to be undernourished. Women from all the divisions were less likely to be undernourished as compared to Sylhet division.

### 3.3. Socioeconomic inequalities of women's undernutrition

Table 3 shows the results of the concentration index used to examine socioeconomic inequalities in undernutrition among ever-married women in urban and rural Bangladesh. The result shows that about 26 percent of the socio-economic inequality among undernourished women (CIX = -0.26) was found in urban residences in Bangladesh, whereas the concentration index value for rural residences in Bangladesh was -0.28. The negative value of CIX indicates that the inequality is more in favor of the poorest households; that is, undernutrition is more concentrated in households with the poorest wealth index than in the richest households. The absolute difference in the distribution of undernutrition between the quantile groups of the poorest and richest wealth was 11.3% for urban residences and 17.1% for rural residences in Bangladesh. Moreover, this study found a 5.2 percent poorest vs. richest ratio for the prevalence of undernutrition among ever-married women in Bangladesh, which was 1.7 percent higher than urban residence.

Fig 2 shows the socioeconomic inequalities among the undernutrition of ever-married women in the divisions of Bangladesh. Fig 2 depicts that Chattogram divisoin shows highest inequality among other divisions of Bangladesh. Approximatey 27 percent inequality in undernourished ever-married women who live in Chattogram division (CIX: −0.27).

**Table 2. Binary logistic regression analysis showing the effects of undernutrition status among ever-married women by background characteristics in urban and rural residence.**

| Background characteristics | Urban | | Rural | |
|---|---|---|---|---|
| | OR | 95% CI | OR | 95% CI |
| **Respondent age (in years)** | | | | |
| 15–19 (ref.) | 1 | | 1 | |
| 20–24 | 0.48*** | 0.35–0.65 | 0.56*** | 0.47–0.68 |
| 25–29 | 0.21*** | 0.15–0.30 | 0.36*** | 0.30–0.44 |
| 30–34 | 0.13*** | 0.09–0.20 | 0.32*** | 0.26–0.39 |
| 35–39 | 0.17*** | 0.11–0.25 | 0.26*** | 0.21–0.32 |
| 40–44 | 0.14*** | 0.09–0.22 | 0.34*** | 0.27–0.42 |
| 45–49 | 0.18*** | 0.12–0.28 | 0.35*** | 0.28–0.44 |
| **Respondent education** | | | | |
| No education (ref.) | 1 | | 1 | |
| Primary | 0.80 | 0.59–1.09 | 0.83* | 0.71–0.96 |
| Secondary | 0.64** | 0.45–0.89 | 0.74** | 0.62–0.87 |
| Higher | 0.60* | 0.39–0.93 | 0.79* | 0.61–1.03 |
| **Respondent working status** | | | | |
| Yes (ref.) | 1 | | 1 | |
| No | 0.90 | 0.73–1.12 | 0.88* | 0.78–0.98 |
| **Marital status** | | | | |
| Others (ref.) | 1 | | 1 | |
| Married | 0.67* | 0.46–0.98 | 0.81* | 0.66–0.99 |
| **Mass media access** | | | | |
| Not exposed (ref.) | 1 | | | |
| Exposed | 0.62*** | 0.49–0.78 | 0.83** | 0.73–0.94 |
| **Wealth status** | | | | |
| Poorest | 2.55*** | 1.64–3.95 | 4.60*** | 3.37–6.28 |
| Poorer | 2.44*** | 1.60–3.71 | 3.68*** | 2.72–4.98 |
| Middle | 2.52*** | 1.79–3.54 | 2.38*** | 1.76–3.22 |
| Richer | 1.90*** | 1.44–2.51 | 1.90*** | 1.39–2.60 |
| Richest (ref.) | 1 | | 1 | |
| **Divisions** | | | | |
| Barisal | 0.33** | 0.16–0.66 | 0.40*** | 0.30–0.52 |
| Chattogram | 0.37*** | 0.23–0.60 | 0.32*** | 0.26–0.40 |
| Dhaka | 0.51** | 0.33–0.79 | 0.47*** | 0.38–0.58 |
| Khulna | 0.49** | 0.29–0.82 | 0.47*** | 0.37–0.59 |
| Mymensingh | 0.45* | 0.25–0.83 | 0.70** | 0.56–0.88 |
| Rajshahi | 0.41** | 0.25–0.70 | 0.52*** | 0.42–0.65 |
| Rangpur | 0.45** | 0.25–0.80 | 0.48*** | 0.39–0.60 |
| Sylhet (ref.) | 1 | | 1 | |

(ref.) = Reference category; Statistically significance

*P<0.05

**P<0.01

***P<0.001.

**Table 3. Regional variations of socioeconomic inequalities in ever-married undernourished women.**

| Inequality indices | Urban | Rural |
|---|---|---|
| Poorest: Richest | 3.5 | 5.2 |
| Poorest–Richest | 11.3 | 17.1 |
| Concentration Index | -0.26*** | -0.28*** |

Significance at

***P<0.001.

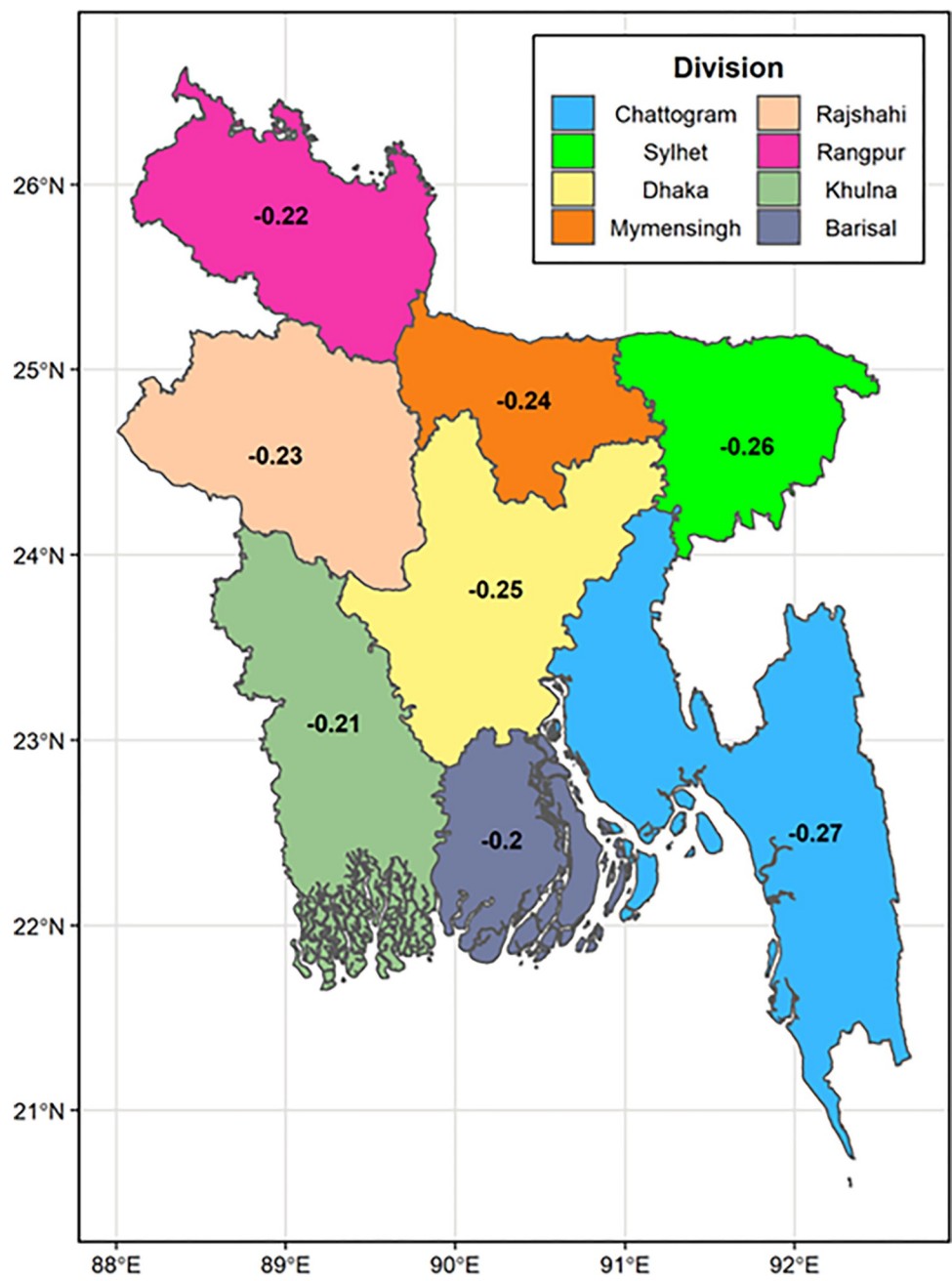

**Fig 2. Socioeconomic inequalities among ever-married women's undernutrition across the divisions of Bangladesh.**

### 3.4. Decomposition of the urban-rural undernutrition gap

The results of Blinder-Oaxaca decomposition of residential undernutrition gap were presented in Table 4. This study evaluates the gap due to the

i.  Difference in the characteristics (E)

ii.  Difference in the effect of the coefficient (C), and

iii.  Interaction (CE)

 The results of Table 4 show that the prevalence probability of being undernourished was 0.091 (approximately 10 percent) for ever married urban women and 0.142 (approximately 15 percent) for ever married rural women in Bangladesh. That is, the prevalence was significantly higher in rural residences compared to urban residences in Bangladesh. There was a significant gap between the urban and rural groups, and the mean difference was 0.051. This difference is decomposed into endowments, coefficients, and interactions. Only the endowment effect was statistically significant, accounting for -0.044 (approximately 86 percent) of the total urban-rural difference in undernutrition prevalence. This means that differences in respondents' endowments explain 86 percent of the differences between urban and rural Bangladesh.

### 3.5. Contribution of each determinant in the urban-rural gap

Table 5 demonstrates the contribution of the endowment and coefficient effects, respectively, in explaining the gap in undernutrition status between urban and rural ever-married women in Bangladesh. A negative contribution indicates that the determinant was narrowing the gap between the urban and rural, and vice versa. According to the difference in covariate distribution, wealth status, working status of the respondents, access to the mass media, and age of the respondents (in years) were the most important contributors to explain the gap between urban and rural residence. For example, the poorest (27 percent) and poorer (29 percent) were the highest contributors maximizing the urban-rural gap. In addition, women's higher education (10 percent) and exposure to the media (15 percent) has contributed a lot to widening the gap. Factors like richer wealth status, older aged women appeared to lessen the gap in undernutrition status between urban-rural residence. Table 5 also shows the part of the gap that was

**Table 4. Blinder-Oaxaca decomposition estimates of undernutrition status by residence among ever-married women.**

|  | Estimate | SE | 95% CI |
|---|---|---|---|
| *Predicted probability* |  |  |  |
|  Urban | 0.091*** | 0.003 | (0.084, 0.098) |
|  Rural | 0.142*** | 0.003 | (0.136, 0.148) |
| *Difference in predicted probability* |  |  |  |
|  Total difference (R) | -0.051*** | 0.005 | (-0.06, -0.042) |
| *Decomposition* |  |  |  |
|  Difference due to endowments (E) | -0.044*** | 0.004 | (-0.053, -0.035) |
|  Difference due to coefficients (C) | 0.002 | 0.005 | (-0.007, 0.012) |
|  Difference due to interaction (CE) | -0.009 | 0.005 | (-0.02, 0.001) |

Significance at

***P<0.001.

**Table 5. Covariates explain most urban-rural gap in undernutrition status for ever-married women in Bangladesh.**

| Variables | Endowments contribution (%) | Coefficient contribution (%) |
|---|---|---|
| **Respondent age (in years)** | | |
| 15–19 (ref.) | | |
| 20–24 | -0.13 | -1.2 |
| 25–29 | 4.37*** | -8.5 |
| 30–34 | 0.93 | -17.2 |
| 35–39 | 0.48 | -5.6 |
| 40–44 | -1.57 | -10.8 |
| 45–49 | -0.46 | -7.7 |
| **Respondent education** | | |
| No education (ref.) | | |
| Primary | -2.13 | -0.8 |
| Secondary | -0.02 | -8.5 |
| Higher | 10.09** | -6.9* |
| **Respondent working status** | | |
| Yes (ref.) | | |
| No | 1.62 | 0.1 |
| **Marital status** | | |
| Others (ref.) | | |
| Married | -0.55* | -6.0 |
| **Mass media access** | | |
| Not exposed (ref.) | | |
| Exposed | 14.94** | -15.1 |
| **Wealth status** | | |
| Poorest | 27.31*** | -3.9* |
| Poorer | 29.18*** | -1.3** |
| Middle | 12.34*** | 1.5** |
| Richer | -9.33*** | 2.9** |
| Richest (ref.) | | |
| **Divisions** | | |
| Barisal | -3.06** | -1.7 |
| Chattogram | -0.03 | 4.4 |
| Dhaka | 13.77*** | 4.5 |
| Khulna | 0.19 | 1.3 |
| Mymensingh | -4.60** | -2.2 |
| Rajshahi | -2.17** | -2.8 |
| Rangpur | -5.12*** | -1.0 |
| Sylhet (ref.) | | |

ref. = Reference Category, Significant at

***P<0.001

**P<0.01

*P<0.05.

accounted by different effect of the determinants (coefficient effect) between urban and rural, where negative contribution indicates that it has more protective effect on ever-married women undernutrition in urban-rural gap.

## 4. Discussion

The findings of this study, derived from a nationally representative cross-sectional survey, reveal the presence of socioeconomic inequalities in the prevalence of undernutrition among ever-married women in Bangladesh. Moreover, this research introduces the Blinder-Oaxaca decomposition method to elucidate the factors contributing to the disparities in undernutrition among ever-married women. By conducting a comprehensive analysis, we investigate the demographic and socioeconomic determinants that underlie the urban-rural disparities among ever-married women.

According to data from the Bangladesh Demographic and Health Survey, 2017–18, the overall prevalence of women's undernutrition in Bangladesh was reported as 12 percent. This finding aligns with a previous study conducted in Bangladesh [11]. Examining the residential aspect, it is evident that the prevalence of undernutrition among ever-married women in Bangladesh is higher in rural residences (13.3 percent) compared to urban residences (8.6 percent). This observation is supported by other studies conducted in Bangladesh [19]. A recent study in Bangladesh highlighted the higher prevalence of undernutrition in rural areas, while overnutrition is more prevalent in urban Bangladesh [20]. Furthermore, studies conducted in Tanzania, Ghana and Botswana have consistently shown higher rates of undernutrition among women residing in rural areas compared to their urban counterparts [21–23]. In a related context, regarding children's nutrition in Bangladesh, rural women were found to have a higher probability of being malnourished than urban children, while the situation was reversed for male children [24].

Of the findings regarding overall and regional socioeconomic inequality highlight a concerning trend favoring the poorest households in Bangladesh, suggesting that women undernutrition is more prevalent among this population. The logistic regression model reinforces these observations, revealing that women with a lower wealth index scores are at a higher risk of experiencing undernutrition. These results indicate the presence of socioeconomic disparities in the prevalence of undernutrition among ever-married women of reproductive age in Bangladesh, consistent with findings observed in other low- and middle-income countries [25]. These conclusions align with a previous study conducted in Bangladesh, which also highlighted the disproportionate vulnerability of women from lower-income households to undernourished compared to their counterparts from rich families [11]. Similar results have been reported in other studies conducted in Tanzania and Ethiopia, with the state of malnutrition decreasing as the wealth index increases [24,26].

The results of this study show that the age of women in Bangladesh was significantly and positively associated with the state of undernutrition in both urban and rural areas. Similarly, to the previous study, our findings indicate a deteriorating trend in the undernutrition of women as age increases [27].

This study revealed that highly educated women had a reduced risk of undernutrition compared to those with lower levels of education. Similar findings have been reported in Bangladesh, where study conducted in the same context indicating that women with secondary or higher education had a lower risk of undernutrition compared to those without a formal education [11]. Previous studies have also documented similar results [28]. Furthermore, various studies have highlighted that many uneducated rural women lack sufficient knowledge about maintaining a balanced diet [29–31]. This lack of information may contribute to the higher prevalence of undernutrition among this population. Additionally, another study demonstrated that a higher level of education among women in the community positively influences their utilization of modern medical health services [32].

This study also showed the association between occupation and the likelihood of undernutrition among rural Bangladeshi women, showing that unemployed women had better health outcomes than employed women. This result was consistent with a previous study conducted in Nigeria [33].

Our study included variables related to women's access to mass media. The results of our study showed that women who used the media were less likely to be underweight. Access to media serves as a readily available source of information for promoting proper nutrition and healthy behaviors.These results are consistent with studies from Botswana and Nepal, which reported that women who had access to the media were less likely to be thin or underweight [23,34]. Regular TV viewing has also been linked to an increased risk of obesity and being overweight. Bangladeshi women's weight increases because they are less physically active and spend more time sitting down [35].

Researchers observed that media use has no effect on body weight in a study of secondary students in Kuching, South Malaysia, because respondents had normal weights and ate fewer harmful foods [36]. The frequency of reading women's magazines was positively correlated with the likelihood of dieting to lose weight due to a magazine article, starting an exercise regimen due to a magazine article, and believing that magazine pictures influence their perception of ideal body shape [37].

The findings indicate that urban households in Bangladesh accounted for around 26% of the socioeconomic inequality among undernourished women, while rural households in Bangladesh had a concentration index value of -0.28. Previous studies estimated and analyzed the levels of socioeconomic inequalities in nutritional status among ever-married women of reproductive age in overall Bangladesh using the concentration index, and they found that undernutrition was more common in Bangladesh's socioeconomically weakest (poorest) group (C = 0.26) [11].

The finding has shown that there is a considerable gap (approximately 5 percent) in women's undernutrition between urban and rural residences. In a previous study based on nutritional inequalities, it was shown that there was a mean BMI gap (total predicted gap) of 4.1 between the poorest and richest women in Bangladesh [36]. Demographic and socioeconomic factors explained 86% of the urban-rural disparity in undernutrition. Almost 40% of the nutritional inequality between two extreme groups of wealth indices can be explained by all predictors and those researchers included in the Blinder-Oaxaca (BO) decomposition model [36]. Among these, wealth status, respondent working status, mass media access, and respondent age (in years) were the most important contributors to explaining the gap. Furthermore, in this different context, wealth status was the most important contributor explaining the difference between urban and rural residence, according to differences in the covariate distribution. For example, the poorest (27 percent) and poorer (29 percent) were the highest contributors, maximizing the urban-rural inequality gap among ever-married women in Bangladesh. The concentration index (CIX) value for inequality in minimum dietary diversity (MDD) among Bangladeshi children aged 6–23 months due to wealth status was positive, and with wealth status accounting for the highest 49.47% of inequality followed by, the mother's education level accounting for 25.06%, and the number of antenatal care (ANC) visits accounting for 20.41% [38].

The main strength of the study is the use of nationally representative data to identify explanatory variables for explaining the urban-rural disparities in women's undernutrition in Bangladesh. To better comprehend the factors driving the observed urban-rural disparities in women's undernutrition, our research employs the Blinder-Oaxaca decomposition technique. This method allows us to disentangle the effects of various factors contributing to the disparities, thereby providing a nuanced understanding of the underlying drivers. However, it is

important to acknowledge the limitations of the study. Firstly, the proportion of the contribution of each category of explanatory variables is not known. Secondly, given the cross-sectional design of the study, establishing causal-effect relationships was not feasible.

## 5. Conclusion

This study employed a combination of statistical analysis, concentration indexing measures, and Blinder-Oaxaca decomposition techniques to explore the factors contributing to the urban-rural variation in women's undernutrition in Bangladesh. We found that about 26% of the socioeconomic inequality among undernourished women was found in Bangladesh. Age, education, marital status, mass media access, wealth status, and division were the most significant risk factors associated with undernutrition in our study population. It was observed that the prevalence was significantly higher in rural residences compared to urban residences in Bangladesh. According to the difference in the covariate distribution, the frequency of wealth status, the working status of the respondent, the access to the media and the age of the respondent were the most important contributors to explaining the gap between urban and rural residences. The findings of this study have important implications for informing the government, policymakers, and other public health stakeholders to mitigate the socioeconomic inequalities in women's undernutrition in Bangladesh. Education and economic status of women close the gender gap in rural undernutrition, potentially promoting maternal health policies in Bangladesh.

## Acknowledgments

The authors thank the Demographic Health Survey for allowing us to use data from the Bangladesh Demographic and Health Survey for this study.

## Author Contributions

**Conceptualization:** Md. Ismail Hossain, Azizur Rahman.

**Data curation:** Md. Jakaria Habib.

**Formal analysis:** Md. Ismail Hossain.

**Methodology:** Md. Ismail Hossain, Md. Jakaria Habib, Azizur Rahman, Md Injamul Haq Methun.

**Supervision:** Md Injamul Haq Methun.

**Writing – original draft:** Md. Ismail Hossain, Md. Jakaria Habib, Faozia Afia Zinia, Iqramul Haq.

**Writing – review & editing:** Faozia Afia Zinia, Azizur Rahman, Md Injamul Haq Methun, Iqramul Haq.

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
