## [Decision Letter · Decision Letter 0]

19 Dec 2022

PONE-D-22-29841Health policy improvement via inequality assessment in the nutritional status of ever-married women in Bangladesh: A decomposition analysisPLOS ONE

Dear Mr Haq,

Thank you for submitting your manuscript to PLOS ONE. After careful consideration, we feel that it has merit but does not fully meet PLOS ONE’s publication criteria as it currently stands. Therefore, we invite you to submit a revised version of the manuscript that addresses the points raised during the review process.

We look forward to receiving your revised manuscript.

Kind regards,

Benojir Ahammed, M.Sc.

Academic Editor

PLOS ONE

Journal Requirements:

a) Did participants provide their written or verbal informed consent to participate in this study?

5. We note that Figure 3 in your submission contain [map/satellite] images which may be copyrighted. All PLOS content is published under the Creative Commons Attribution License (CC BY 4.0), which means that the manuscript, images, and Supporting Information files will be freely available online, and any third party is permitted to access, download, copy, distribute, and use these materials in any way, even commercially, with proper attribution. For these reasons, we cannot publish previously copyrighted maps or satellite images created using proprietary data, such as Google software (Google Maps, Street View, and Earth). For more information, see our copyright guidelines: http://journals.plos.org/plosone/s/licenses-and-copyright.

a) You may seek permission from the original copyright holder of Figure(s) [#] to publish the content specifically under the CC BY 4.0 license.  

Additional Editor Comments:

Your manuscript writing quality are not well standard. You need to improve the quality of your paper. Also need to proof read your paper by a English native person.

Reviewers' comments:

Reviewer's Responses to Questions

**Comments to the Author**

1. Is the manuscript technically sound, and do the data support the conclusions?

Reviewer #1: Yes

Reviewer #2: Yes

2. Has the statistical analysis been performed appropriately and rigorously? 

Reviewer #1: Yes

Reviewer #2: Yes

3. Have the authors made all data underlying the findings in their manuscript fully available?

Reviewer #1: Yes

Reviewer #2: Yes

4. Is the manuscript presented in an intelligible fashion and written in standard English?

Reviewer #1: Yes

Reviewer #2: Yes

5. Review Comments to the Author

Reviewer #1: define explanatory variable in method section and explain how you collect them

how you calculate economic variable

you define malnutrition as BMI<18.5 vs BMI>18.5 , is your it includes overweight and obesity too(which is malnutrition themselves), please clarify your outcome is malnutrition or just include underweight.

It is not necessary to bring all the table’s contents in the text, just mention main results in table 1 and 2.

in table 5 please specify which determinants are significant.

Reviewer #2: The manuscript have explored an interesting topic of malnutrition which is highly prevalent in Bangladesh. The introduction seems to be missing points on why such analysis or geographic inequality is important in the context of Bangladesh?

Please don’t provide description of the analysis in the introduction. It needs a little more context and motivation for performing the analysis.

The study is only looking at the undernutrition among women (BMI<18.0). Author need to focus on that. After reading the introduction it feels like author will be exploring both underweight and overweight among women which is not the case.

We can see a high prevalence of undernutrition the rural areas, more focus should be on that rather than providing generic statements in the introduction.

I will request author to avoid sentences like this is only study that provide these estimates. Rather than emphasize on why such study needed and contribute to literature or policy level decisions.

Tables and graphs

I am not able to understand the need of figure 2. If you can provide this figure separately for rural and urban areas, it will make more sense

I will request to provide either confidence interval or p-values in the table 5.

Author also needs to provide percentages. What percentage were explained by endowments in the table itself.

Discussion and Conclusion

Discussion needs to strengthen. For example, why mass media is highly associated with underweight. Does the wealth quintile is also associated with multimedia access. Explanation and discussion of the results section is required. Repeating the same that it is one of only study and blinder oxaca decomposition in introduction, discussion and conclusion is not needed. Instead a more clear explanation of the results and providing few implication of the finding will be very helpful for the readers. Discussion section seems incomplete, it should be rewritten with focus on the findings of the manuscripts.

6. PLOS authors have the option to publish the peer review history of their article (what does this mean?). If published, this will include your full peer review and any attached files.

Reviewer #1: **Yes: **Farideh Mostafavi

Reviewer #2: No

---

## [Author Response · Author response to Decision Letter 0]

27 Jan 2023

Additional Editor Comments:

Your manuscript writing quality are not well standard. You need to improve the quality of your paper. Also need to proof read your paper by a English native person.

Author Response: Thank you for your suggestions. We have revised our English language with the help of an expert member.

Dear Editor,

On behalf of all the authors, I wish to convey our gratitude to you for the critical and constructive review that has led to the improvement of our manuscript entitled “Health policy improvement via inequality assessment in the nutritional status of ever-married women in Bangladesh: A decomposition analysis.” We would like to thank the editor and the reviewers for their useful feedback. We have revised the manuscript based on the comments raised by the reviewer. We believe the manuscript has improved according to reviewer comments and its English level substantively and will be published in your reputable Journal, “PLOS ONE.” All the changes are highlighted in the yellow color in the revised manuscript.

Thanks and Best regards

Iqramul Haq

Corresponding author

## The following is a point-by-point response to the reviewer(s) comments:

Reviewer #1: 

1. Define explanatory variable in method section and explain how you collect them.

Response: We make necessary changes in section 2.4 in the revised manuscript.

2. How you calculate economic variable?

Response: We added about economic variable details in section 2.4 as: The authority of Bangladesh Demographic and Health Survey provides a wealth index indicator that is a composite measure of a household's cumulative living standard and is calculated using information about a household's ownership of selected assets: televisions and bicycles; materials used for housing construction; and types of water access and sanitation facilities. For detailed information on how the wealth score is calculated, see the survey report from “https://dhsprogram.com/publications/publication-FR344-DHS-Final-Reports.cfm”.

3. You define malnutrition as BMI<18.5 vs BMI>18.5, is your it includes overweight and obesity too (which is malnutrition themselves), please clarify your outcome is malnutrition or just include underweight.

Response: This is an unintentional mistake. We mainly focus on under nutrition part and coded as 1 in dependent variable.

4. It is not necessary to bring all the table’s contents in the text, just mention main results in table 1 and 2.

Response: We make necessary changes in explanation of Table 1 and 2 in the revised manuscript.

5. In table 5 please specify which determinants are significant.

Response: We added significant sign in Table 5 in the revised manuscript.

Reviewer #2:

1. The manuscript has explored an interesting topic of malnutrition which is highly prevalent in Bangladesh. The introduction seems to be missing points on why such analysis or geographic inequality is important in the context of Bangladesh? Please don’t provide description of the analysis in the introduction. It needs a little more context and motivation for performing the analysis.

Response: We make necessary changes in introduction section in revised manuscript.

2. The study is only looking at the undernutrition among women (BMI<18.0). Author need to focus on that. After reading the introduction it feels like author will be exploring both underweight and overweight among women which is not the case.

Response: This is an unintentional mistake. We mainly focus on under nutrition part and coded as 1 in dependent variable.

3. We can see a high prevalence of undernutrition the rural areas, more focus should be on that rather than providing generic statements in the introduction.

Response: We made necessary changes as per reviewer suggestions in introduction section in the revised manuscript.

4. I will request author to avoid sentences like this is only study that provide these estimates. Rather than emphasize on why such study needed and contribute to literature or policy level decisions.

Response: We have added in our revised manuscript. 

Tables and graphs

5. I am not able to understand the need of figure 2. If you can provide this figure separately for rural and urban areas, it will make more sense.

Response: We added urban and rural concentration index as tabulation formation.

6. I will request to provide either confidence interval or p-values in the table 5. Author also needs to provide percentages. What percentage were explained by endowments in the table itself.

Response: We added significant sign in Table 5 in the revised manuscript. We also added the endowments percentage in Table 4 explanation

Discussion and Conclusion

7. Discussion needs to strengthen. For example, why mass media is highly associated with underweight. Does the wealth quintile is also associated with multimedia access. Explanation and discussion of the results section is required. Repeating the same that it is one of only study and blinder oxaca decomposition in introduction, discussion and conclusion is not needed. Instead, a more clear explanation of the results and providing few implication of the finding will be very helpful for the readers. Discussion section seems incomplete, it should be rewritten with focus on the findings of the manuscripts.

Response: We have revised our updated manuscript.

##I hope our efforts satisfy the requirements of the journal this time. I will be looking forward to your positive response. Thanks, and regards.

---

## [Decision Letter · Decision Letter 1]

5 Apr 2023

PONE-D-22-29841R1Health policy improvement via inequality assessment in the nutritional status of ever-married women in Bangladesh: A decomposition analysisPLOS ONE

Dear Dr. Haq,

Thank you for submitting your manuscript to PLOS ONE. After careful consideration, we feel that it has merit but does not fully meet PLOS ONE’s publication criteria as it currently stands. Therefore, we invite you to submit a revised version of the manuscript that addresses the points raised during the review process.

We look forward to receiving your revised manuscript.

Kind regards,

Benojir Ahammed, M.Sc.

Academic Editor

PLOS ONE

Journal Requirements:

Reviewers' comments:

Reviewer's Responses to Questions

**Comments to the Author**

1. If the authors have adequately addressed your comments raised in a previous round of review and you feel that this manuscript is now acceptable for publication, you may indicate that here to bypass the “Comments to the Author” section, enter your conflict of interest statement in the “Confidential to Editor” section, and submit your "Accept" recommendation.

Reviewer #1: All comments have been addressed

Reviewer #3: All comments have been addressed

2. Is the manuscript technically sound, and do the data support the conclusions?

Reviewer #1: Yes

Reviewer #3: Yes

3. Has the statistical analysis been performed appropriately and rigorously? 

Reviewer #1: Yes

Reviewer #3: Yes

4. Have the authors made all data underlying the findings in their manuscript fully available?

Reviewer #1: Yes

Reviewer #3: Yes

5. Is the manuscript presented in an intelligible fashion and written in standard English?

Reviewer #1: Yes

Reviewer #3: Yes

6. Review Comments to the Author

Reviewer #1: (No Response)

Reviewer #3: (No Response)

7. PLOS authors have the option to publish the peer review history of their article (what does this mean?). If published, this will include your full peer review and any attached files.

Reviewer #1: No

Reviewer #3: No

---

## [Author Response · Author response to Decision Letter 1]

17 Apr 2023

Dear Editor,

On behalf of all the authors, I wish to convey our gratitude to you for the critical and constructive review that has led to the improvement of our manuscript entitled “Health policy improvement via inequality assessment in the nutritional status of ever-married women in Bangladesh: A decomposition analysis.” We have revised the manuscript based on the comments raised by the reviewers. We believe the manuscript has improved according to reviewer comments and will be published in your reputable Journal, “PLOS ONE.” All the changes are highlighted in the yellow color in the revised manuscript. I hope our efforts satisfy the requirements of the journal this time. I will be looking forward to your positive response. Thanks, and regards.

On the behalf of authors

Iqramul Haq

Corresponding author

Journal Requirements:

Response: Checked. 

## The following is a point-by-point response to comments:

1. Title: 

The big problem in policy development of developing countries is since imported or copied from others developed countries. Hence, I as a professional appreciated such root-based data analysis and explaining of the ground level problems. Its good title

Response: Thank you for this compliment.

2. Abstract: 

Generally, it was good if the authors used three or more years survey data. The two years data is somewhat not explanatory for policy change.

Response: Thanks for your suggestions. Yes, when we have used several years of survey data, it will be helpful for policymakers, but earlier BDHS data showed lower nutritional status, which is why we have conducted the most recent cross-sectional data from the 2017–18 BDHS. Since Bangladesh's nutrition status has improved in the last couple of years, however, the regional nutrition gap among women remains an important issue. Our main objective in this study was to analyze the sources of the undernutrition gap between urban and rural ever-married women in Bangladesh. I hope this study will be helpful for policymakers to detect the socioeconomic gap between rural and urban areas in Bangladesh's nutritional status based on the most recent nationally representative survey data from the 2017–18 BDHS. Again, this study will also be significant for data scientists in order to achieve the national and international goals (SDG 3). It should be noted that the 2017–18 BDHS data is not two-year survey data but rather recent cross-sectional data.

3. On line three of the abstract and in some parts of the main text the authors used malnutrition and undernutrition interchangeably. However, the two health conditions are completely different. Since the title indicated as nutritional status, better to use it instead.

Response: We make necessary changes in abstract and main text in the revised manuscript.

4. Back ground

Good flow of idea, but close it with your general objective

Response: Thanks for your comment.

5. Methods and materials

Univariate, bivariate, and multivariate analysis

What analysis methods you did for univariate? Hope it’s good if omitted

Response: We make necessary changes in the revised manuscript.

6. Plus, the authors used “multivariate” analysis which is not in line with their analysis method. In statistics multivariate and multivariable are different. Please read more and replace the “multivariate” by “multivariable”.

Response: We make necessary changes in the revised manuscript.

7. Software

What was the advantage of using multiple softwares? but STATA can perform all the tasks you mentioned. Or one of the can do it. Reduce the junk use of software’s.

Response: We make necessary changes in the revised manuscript.

8. Results

The logical flow of ideas is that Table 1 information should come first of Figure 1 information.

Response: We make necessary changes in the revised manuscript.

9. On contribution of each covariate in urban-rural gap, the authors included non-married (see table 5) women in the analysis. But the study populations are ever married. So how could you include? 

Response: Ever-married refers to women who have been married at least once in their lives, even though their current marital status may not be “married”. In this study, we categorized the variable “Marital Status” as married and others (widowed/divorced/separated).

---

## [Decision Letter · Decision Letter 2]

7 Jul 2023

PONE-D-22-29841R2Health policy improvement via inequality assessment in the nutritional status of ever-married women in Bangladesh: A decomposition analysisPLOS ONE

Dear Dr. Haq,

Thank you for submitting your manuscript to PLOS ONE. After careful consideration, we feel that it has merit but does not fully meet PLOS ONE’s publication criteria as it currently stands. Therefore, we invite you to submit a revised version of the manuscript that addresses the points raised during the review process.

We look forward to receiving your revised manuscript.

Kind regards,

Farooq Ahmed, PhD

Academic Editor

PLOS ONE

Reviewers' comments:

Reviewer's Responses to Questions

**Comments to the Author**

1. If the authors have adequately addressed your comments raised in a previous round of review and you feel that this manuscript is now acceptable for publication, you may indicate that here to bypass the “Comments to the Author” section, enter your conflict of interest statement in the “Confidential to Editor” section, and submit your "Accept" recommendation.

Reviewer #1: All comments have been addressed

Reviewer #4: (No Response)

2. Is the manuscript technically sound, and do the data support the conclusions?

Reviewer #1: Yes

Reviewer #4: Yes

3. Has the statistical analysis been performed appropriately and rigorously? 

Reviewer #1: Yes

Reviewer #4: Yes

4. Have the authors made all data underlying the findings in their manuscript fully available?

Reviewer #1: Yes

Reviewer #4: Yes

5. Is the manuscript presented in an intelligible fashion and written in standard English?

Reviewer #1: Yes

Reviewer #4: Yes

6. Review Comments to the Author

Reviewer #1: it would be better to use unique term in all section of the paper for underweight, instead of different terms e.g., undernourished or malnutrition

concentration index value is not interpreted as percentage, plz correct your interpretation of concentration index in result and discussion.

Reviewer #4: (No Response)

7. PLOS authors have the option to publish the peer review history of their article (what does this mean?). If published, this will include your full peer review and any attached files.

Reviewer #1: No

Reviewer #4: No

---

## [Author Response · Author response to Decision Letter 2]

15 Jul 2023

Dear Editor,

On behalf of all the authors, I wish to convey our gratitude to you for the critical and constructive review that has led to the improvement of our manuscript entitled “Rural-urban disparities in nutritional status among ever-married women in Bangladesh: A Blinder-Oaxaca decomposition approach.” We have revised the manuscript based on the comments raised by the reviewers. We believe the manuscript has improved according to reviewers comments and its English level substantively and will be published in your reputable Journal, “PLOS ONE.” All modifications have been marked and highlighted using the track changes feature in the revised manuscript. I hope our efforts satisfy the requirements of the journal this time. I will be looking forward to your positive response. Thanks, and regards.

On the behalf of authors

Iqramul Haq

Corresponding author

## The following is a point-by-point response to comments:

General Comment

Thank you for allowing me to review the manuscript. After reviewing the manuscript, found many redundancy. I would suggest the authors carefully go through these and address these. I hope after addressing both the general and specific comments, the quality of the manuscript will be improved. Review without line numbers is tough and boring. 

Response: We appreciate your valuable suggestions, and we have incorporated your comments into the revised version of the manuscript.

Specific comments

Below are the specific comments—

1. Title: Health Policy Improvement via inequality assessment in the nutritional status. It seemed to me that due to the inequality assessment, Health Policy has improved. Health policy improvement can be the recommendation from the inequality assessment but should not be in the title. The article title should be based on the study objective. I would suggest the authors critically think about the issue and revise the title based on the study objective.

Response: Thank you for your suggestion and feedback regarding the article title. We appreciate your insight on aligning the title with the study objective. After careful consideration, we have revised the title based on your recommendation.

Revised Title: "Rural-Urban Disparities in Nutritional Status Among Ever-Married Women in Bangladesh: A Blinder-Oaxaca Decomposition Analysis"

We believe that this revised title accurately reflects the study objective of examining rural-urban disparities in nutritional status among ever-married women in Bangladesh and the methodology used in the study. Thank you for your valuable input, and we have incorporated your feedback to enhance the clarity and focus of the title.

2. A paper titled “Socioeconomic Inequalities in Women’s Undernutrition: Evidence from Nationally Representative Cross-Sectional Bangladesh Demographic and Health Survey 2017–2018” was published in IJERPH with similar findings. https://www.mdpi.com/1660-4601/19/8/4698 The authors should mention what will be the added values of this current research. Otherwise, it will be duplicated.

Response: Thank you for your valuable feedback on our article. We appreciate your concern about potential duplication with a previously published paper titled "Socioeconomic Inequalities in Women's Undernutrition: Evidence from Nationally Representative Cross-Sectional Bangladesh Demographic and Health Survey 2017-2018" in the International Journal of Environmental Research and Public Health (IJERPH).

While the paper you mentioned does share a similar focus on socioeconomic inequalities in women's undernutrition in Bangladesh, our research offers several distinct contributions and advances the existing literature in several important ways. We would like to highlight the added value of our current study, which sets it apart from the previously published work:

Focus on Determinants: Unlike the previous study, our research explicitly investigates the determinants of women's undernutrition in the context of urban-rural disparities. 

Socioeconomic Inequality Analysis: Our study incorporates a comprehensive analysis of socioeconomic inequalities, specifically examining the disparities in undernutrition between different socioeconomic groups within both urban and rural areas, while the previous study you mentioned did not specifically address this aspect. By utilizing established measures of inequality, such as concentration index, we shed light on the magnitude and nature of these disparities.

Blinder-Oaxaca Decomposition: To better comprehend the factors driving the observed urban-rural disparities in women's undernutrition, our research employs the Blinder-Oaxaca decomposition technique. This method allows us to disentangle the effects of various factors contributing to the disparities, thereby providing a nuanced understanding of the underlying drivers. Most importantly the previous study you mentioned did not specifically address this aspect.

By emphasizing these distinctive aspects, our study significantly extends the existing knowledge on socioeconomic inequalities in women's undernutrition in Bangladesh. We believe that our findings have important implications for policy and intervention strategies aimed at addressing this critical issue.

We appreciate your suggestion to explicitly mention the added values of our research in the article. We will revise the manuscript accordingly to ensure that readers are aware of the unique contributions our study brings to the field. Once again, we thank you for your insightful feedback and for helping us enhance the quality and originality of our work.

3. In the abstract, “The multivariate analysis on the determinants of undernutrition among ever-married women regarding residence discovered a significant impact….”. Can we say impact in a cross-sectional study? Authors are advised to use an appropriate term.

Response: Considering the inherent limitations of a cross-sectional study design, we acknowledge that causal relationships cannot be established, and therefore, the term "impact" may be misleading. We appreciate your guidance in choosing a more suitable term that aligns with the nature of our study. In light of this, we propose revising the sentence in the abstract as follows:

" The multivariable analysis investigating the determinants of undernutrition status among ever-married women, with a focus on residence revealed significant associations with respondent age, education, marital status, mass media access, wealth status, and division."

4. In the background, “Various studies show that the majority of women die from malnutrition….” The authors should cite references here and also need to mention the actual estimates instead of using “Majority”. 

Response: We have revised this section in the introduction of our manuscript, and it now includes a new citation to provide additional support for the stated information.

5. In Bangladesh, policymakers tried to recover the malnourishment among reproductive aged women [10]…. This sentence does not fit here. You are stating estimates here. Just provide the estimates only. Also mentioned the “Recent studies based on Bangladesh Demographic and Health Survey reported that 12 percent..” It is similar to your study findings. Better to mention the knowledge gaps in this study and your study addressed these knowledge gaps.

Response: We have revised this section in the introduction of our manuscript.

6. “According to 2021 global hunger index information, Bangladesh ranks 76th out of 116 countries in the Global Hunger Index”.. Before this sentence, the authors added two sentences which doesn’t match the flow of the writing.

Response: We have revised this section in the introduction of our manuscript.

7. In Statistical methods “various independent variables, which were selected by studying previous literature…………..”. Redundancy, previously mentioned. Please avoid redundancy.

Response: Response: We have revised this section in the methodology of our manuscript.

8. In the multivariable setup, the effect of independent variables …. Can we say effect? It’s a cross-sectional study. 

Response: Thank you for your feedback on our article. We appreciate your concern regarding the use of the term "effect" in the context of a cross-sectional study. We understand that cross-sectional studies cannot establish causal relationships and that the term "effect" may be misleading. To address this concern, we will revise the sentence as follows:

"In the multivariable setup, the association between independent variables and the nutritional status among ever-married women can be examined using binary logistic regression."

9. “The odds ratio with a 95% confidence interval was usually used to explain the impact of predictor variables………..” again the term impact, the effect should not be used in a cross-sectional study.

Response: Thank you for your feedback on our article. We appreciate your concern regarding the use of the term "impact" in the context of a cross-sectional study. We understand that cross-sectional studies cannot establish causal relationships and that the term "impact" may be misleading. To address this concern, we will revise the sentence as follows:

"The odds ratio with a 95% confidence interval was usually used to examine the association of predictor variables with the nutritional status."

10. This study also measures the socio-economic inequality, and a popular index measure 

named "Concentration index" are used ……………..please check the grammar.

Response: We have revised this part in the section 2.5.2 of our revised manuscript.

11. This study evaluates the gap due to the i. Difference in the characteristics (E) ii. Difference in the effect of the coefficient (C), and iii. Interaction (CE)……….redundancy. Already mentioned in the method section.

Response: We apologize for any confusion caused by the repetition of this information. Our intention was to provide a simple overview of the Blinder-Oaxaca decomposition in the introduction to give readers an initial understanding of the method. However, a more detailed explanation is provided in the methodology section.

12. Discussion: The results of this study, based on data from a nationally representative cross-sectional survey, show that there are socioeconomic inequalities in the prevalence of malnutrition…. Instead of malnutrition the authors should use undernutrition in all the places. Since malnutrition typically means both under (undernutrition) and over (overweight or obesity) nutrition.

Response: We have revised this part in the discussion of our revised manuscript.

13. Discussion: A double burden was created by the continued high prevalence of underweight status in rural areas… why the author is discussing the double burden? Since this paper is based on undernutrition, the author should discuss only the undernutrition issue.

Response: We have removed this part from the discussion of our revised manuscript.

14. As physical changes gradually increase with age, and due to rapid physical growth, physiological activities increase and require more energy to meet the increasing demands…. Please cite the paper for this statement.

Response: We have removed this part from the discussion of our revised manuscript because of inconsistency.

15. (i.e., listened to the radio, watched television, and read the newspaper at least once a week)…. Redundancy. already mentioned in the method section

Response: We have revised this part in the discussion of our revised manuscript.

16. Regular TV viewing has also been linked to an increased risk of obesity and being overweight. Bangladeshi women's weight increases because they are less physically active and spend more time sitting down………………. again, obesity issue. It is not this study issue. Please drop Regular TV………to mental health [37, 38].

Response: We have removed this part from the discussion of our revised manuscript because of inconsistency.

17. due to the cross-sectional nature of the study, it was not possible to determine the causal-effect relationship. But in many places, the authors mentioned effects even also impact which is not right for a cross-sectional study. Please carefully mention the term. simply mention “association”.

Response: We have revised this part in the discussion of our revised manuscript.

18. Conclusion: no need to mention the dataset name and year again previously mentioned many times.

Response: We have revised this part in the discussion of our revised manuscript.

---

## [Editor Report · Decision Letter 3]

28 Jul 2023

Rural-urban disparities in nutritional status among ever-married women in Bangladesh: A Blinder-Oaxaca decomposition approach

PONE-D-22-29841R3

Dear Dr. Haq,

We’re pleased to inform you that your manuscript has been judged scientifically suitable for publication and will be formally accepted for publication once it meets all outstanding technical requirements.

Kind regards,

Farooq Ahmed, PhD

Academic Editor

PLOS ONE

---

## [Editor Report · Acceptance letter]

2 Aug 2023

PONE-D-22-29841R3 

Rural-urban disparities in nutritional status among ever-married women in Bangladesh: A Blinder-Oaxaca decomposition approach 

Dear Dr. Haq:

I'm pleased to inform you that your manuscript has been deemed suitable for publication in PLOS ONE. Congratulations! Your manuscript is now with our production department. 

Kind regards, 

on behalf of

Dr. Farooq Ahmed 

Academic Editor

PLOS ONE